# GAPrompt: Geometry-Aware Point Cloud Prompt for 3D Vision Model

Zixiang Ai [1]   Zichen Liu [1]   Yuanhang Lei [2]   Zhenyu Cui [1]   Xu Zou [3]   Jiahuan Zhou * [1]

## Abstract

Pre-trained 3D vision models have gained significant attention for their promising performance on point cloud data. However, fully fine-tuning these models for downstream tasks is computationally expensive and storage-intensive. Existing parameter-efficient fine-tuning (PEFT) approaches, which focus primarily on input token prompting, struggle to achieve competitive performance due to their limited ability to capture the geometric information inherent in point clouds. To address this challenge, we propose a novel Geometry-Aware Point Cloud Prompt (GAPrompt) that leverages geometric cues to enhance the adaptability of 3D vision models. First, we introduce a Point Prompt that serves as an auxiliary input alongside the original point cloud, explicitly guiding the model to capture fine-grained geometric details. Additionally, we present a Point Shift Prompter designed to extract global shape information from the point cloud, enabling instance-specific geometric adjustments at the input level. Moreover, our proposed Prompt Propagation mechanism incorporates the shape information into the model's feature extraction process, further strengthening its ability to capture essential geometric characteristics. Extensive experiments demonstrate that GAPrompt significantly outperforms state-of-the-art PEFT methods and achieves competitive results compared to full fine-tuning on various benchmarks, while utilizing only 2.19% of trainable parameters. Our code is available at `https://github.com/zhoujiahuan1991/ICML2025-GAPrompt`.

[1]Wangxuan Institute of Computer Technology, Peking University, Beijing, China [2]State Key Laboratory of CAD&CG, Zhejiang University, Hangzhou, China [3]School of Artificial Intelligence and Automation, Huazhong University of Science and Technology, Wuhan, China. Correspondence to: Jiahuan Zhou <jiahuanzhou@pku.edu.cn>.

*Proceedings of the 42nd International Conference on Machine Learning*, Vancouver, Canada. PMLR 267, 2025. Copyright 2025 by the author(s).

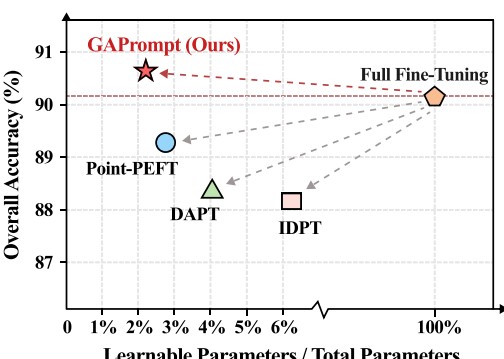

Figure 1. Our GAPrompt compares to full fine-tuning and existing PEFT methods. We compare the classification accuracy on the hardest variant of ScanObjectNN (Uy et al., 2019) based on pre-trained Point-FEMAE (Zha et al., 2024).

## 1. Introduction

The advent of scanning sensor devices has significantly facilitated the acquisition of 3D point cloud data, an inherently irregular and unstructured geometric representation. This advancement has propelled the development of various 3D vision applications, including 3D reconstruction (Xu et al., 2022; Lu et al., 2024) and autonomous driving (Zhao et al., 2024). Recently, pre-trained 3D vision models (Yu et al., 2022; Zhang et al., 2022; Zha et al., 2024) have shown remarkable performance in processing point cloud data, enabling their direct application to a variety of downstream 3D tasks through full fine-tuning. However, fine-tuning the entire pre-trained model incurs substantial computational costs and necessitates a large quantity of labeled data. Moreover, without freezing the pre-trained model, there exists a considerable risk of catastrophic forgetting, potentially resulting in the loss of critical pre-trained knowledge.

To address these challenges, parameter-efficient fine-tuning (PEFT) methods have been introduced, particularly in 2D vision, to improve the efficiency and effectiveness of adapting pre-trained models. The core concept behind PEFT is to freeze the pre-trained model and only fine-tune newly added modules, thereby bridging the distribution gap between pre-training tasks and downstream tasks while preserving the original knowledge. Two notable PEFT strategies are Prompt Tuning (Lester et al., 2021; Li & Liang, 2021; Liu et al., 2024) and Adapter Tuning (Houlsby et al., 2019;

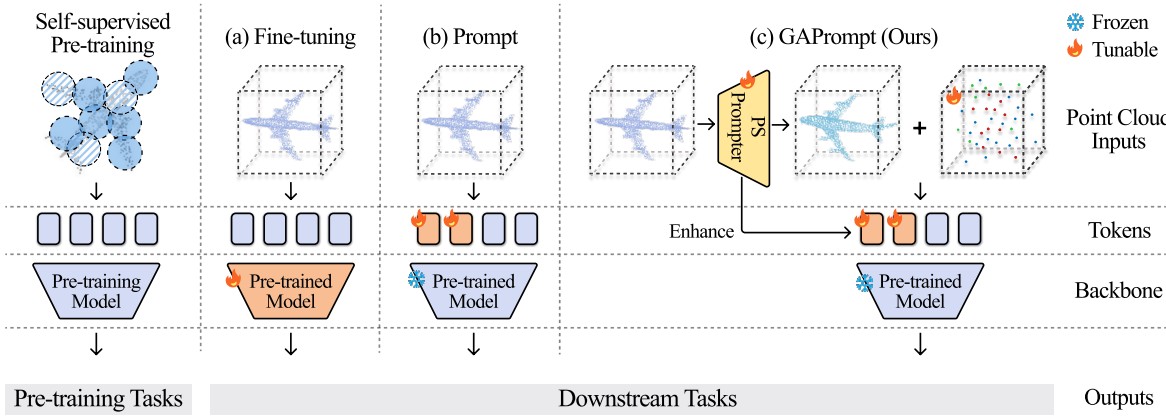

Figure 2. Methods for adapting pre-trained 3D vision models. (a) Fine-tuning updates entire model parameters. (b) Prompt-based methods adapt the model to downstream tasks by reformulating the input at the token level. (c) Our proposed GAPrompt adapts the model by tuning explicit learnable point clouds and token prompts, enhanced with instance-specific shape features extracted by a geometry-aware prompter.

He et al., 2021). However, the transition of these PEFT methods from 2D to 3D vision poses significant challenges due to the inherent sparsity and irregularity of point clouds. Specifically, token prompts initialized randomly often fail to align well with point cloud data, leading to difficulties in convergence when downstream tasks are supervised solely by prediction loss. Similarly, Adapter Tuning, which primarily focuses on token features, struggles to capture the critical geometric information embedded in the distribution of discrete points within 3D space.

Recently, several studies (Zha et al., 2023; Zhou et al., 2024) have recognized the critical need for 3D-specific PEFT methods and have initially tried to design networks that are better suited for adaptation within the 3D domain. These approaches typically focus on either constructing complex networks to model token interactions and dynamically generate token prompts, or employing dynamic adapters that simultaneously produce both prompt tokens and scaling factors for adapter tuning. However, by concentrating primarily on encoded input tokens, these methods fail to capture the rich geometric information intrinsic to point clouds, which severely limits their ability to achieve competitive performance, as illustrated in Figure 1.

To this end, we propose a novel Geometry-Aware Point Cloud Prompt (**GAPrompt**), specifically designed for parameter-efficient fine-tuning of 3D models. Our approach begins with the introduction of a Point Prompt, which explicitly incorporates point cloud data as input, allowing the model to better capture subtle geometric features. To further enhance the capacity of leveraging instance-specific geometry information, we introduce a Point Shift Prompter. This module extracts global shape information from the original point cloud and shifts the points accordingly, thereby enriching the geometric features at the input level. In addition,

we propose a Prompt Propagation mechanism, which seamlessly integrates the instance shape information extracted by the Point Shift Prompter into the model's feature extraction process. This design enhances the model's ability to capture and utilize critical geometric information from the point cloud, leading to more accurate and efficient processing.

In summary, the key contributions of this work are: (1) We propose GAPrompt, a novel geometry-aware prompt learning method tailored for pre-trained 3D vision models. GAPrompt achieves competitive performance comparable to full fine-tuning, while significantly reducing computational and storage overhead. (2) We introduce three key algorithm designs including Point Prompt, Point Shift Prompter, and the Prompt Propagation mechanism, which together enable the model to effectively capture and utilize geometric information inherent in point clouds, thereby enhancing its representational capacity. (3) Extensive experiments on various benchmarks demonstrate the superior efficiency and effectiveness of GAPrompt, outperforming existing methods in both accuracy and resource utilization.

## 2. Related Work

### 2.1. Pre-trained 3D Vision Model

Pre-training on 3D datasets has become a prominent research area, particularly with the use of vision transformers (Dosovitskiy et al., 2021). Two principal pretext task paradigms have been developed for 3D pre-training: contrastive learning and mask modeling. Methods based on contrastive learning (Huang et al., 2023; Dong et al., 2022; Qi et al., 2024; Zhu et al., 2023) have demonstrated remarkable performance in zero/few-shot learning, largely due to the inherent power of multi-modality. Mask modeling (Yu et al., 2022; Zha et al., 2024) typically relies on

autoencoders to learn the latent features by reconstructing the original input as illustrated in Figure 2. For instance, Point-MAE (Pang et al., 2022) employs an autoencoder to learn high-level latent features from unmasked patches and reconstruct the masked point patches. ReCon (Qi et al., 2023) utilizes an ensemble distillation approach, drawing upon both generative modeling teachers and single/cross-modal contrastive teachers. These pre-trained 3D vision models have demonstrated exceptional capabilities when fully fine-tuned for various downstream 3D tasks.

While full fine-tuning can yield promising performance, it is computationally expensive and inefficient, as it requires updating the entire backbone of the model. This has motivated the exploration of more efficient methods for adapting pre-trained models to downstream tasks with minimal parameter updates. In this work, we address this challenge by proposing a geometry-aware prompting approach that leverages instance-specific shape information extracted from input point clouds, enabling efficient transfer of pre-trained models to downstream tasks with significantly fewer trainable parameters.

## 2.2. Parameter-Efficient Fine-Tuning

As deep learning technology advances, both the performance and size of models have steadily increased, making full fine-tuning of these models for downstream tasks computationally expensive. To address these challenges, researchers in 2D computer vision have developed various Parameter-Efficient Fine-Tuning (PEFT) methods. One popular approach, prompt tuning (Jia et al., 2022; Li & Zhou, 2025), introduces learnable latent tokens as prompts to task-specific inputs, allowing pre-trained models to adapt within the latent feature space. Adapter tuning methods (Houlsby et al., 2019) complement this by inserting lightweight modules into the pre-trained model blocks and adjusting the latent feature distribution. Building on these foundations, numerous variations (Jie & Deng, 2023; Karimi Mahabadi et al., 2021) have been proposed, achieving performance levels comparable to full fine-tuning in 2D vision tasks.

However, due to the inherent sparsity and irregularity of point clouds, these 2D vision PEFT methods struggle to generalize effectively to 3D vision tasks, underscoring the need for 3D-specific approaches. For example, IDPT (Zha et al., 2023) utilized a heavy EdgeConv (Phan et al., 2018) network to capture local interactions between tokens, dynamically generating token prompts. While this method narrows the performance gap with full fine-tuning, it significantly increases computational costs. DAPT (Zhou et al., 2024) introduced dynamic adapters for transfer learning in 3D vision models, extending standard adapters to produce both scale factors and prompts dynamically. Additionally, Point-PEFT (Tang et al., 2024) combined prompts, adapters,

and bias tuning (Zaken et al., 2021) to transfer pre-trained models. Nevertheless, it still relies on point priors extracted from pre-training data, further adding computational overhead.

While these pioneering approaches have made significant strides, they often lack an explicit understanding of the geometric characteristics inherent to point clouds, which limits their overall performance. To address this, we propose a novel 3D-specific PEFT method with geometry awareness, termed GAPrompt. Our approach leverages a lightweight prompter to extract instance-specific shape information from point clouds. Furthermore, we integrate explicit learnable point cloud prompts with token prompts to enhance the model's geometric awareness and improve its ability to capture subtle geometric details, as illustrated in Figure 2.

## 3. The Proposed Method

We illustrate our GAPrompt method for efficiently fine-tuning pre-trained 3D vision models in detail, which contains three geometry-aware prompt modules: the Point Prompt, the Point Shift Prompter, and the Prompt Propagation mechanism. As shown in Figure 3, given a pre-trained 3D transformer with $N$ blocks and a specific downstream task, we freeze the backbone and solely update newly introduced GAPrompt modules and the classification head.

### 3.1. Point Prompt

To facilitate explicit awareness of point clouds and assist models in capturing subtle geometry, we design the Point Prompt which explicitly utilizes learnable point clouds as prompt. It has stronger inherent relevance with point cloud data than prompt tokens, assisting models in capturing geometry from inherently irregular and unstructured point clouds. The Point Prompt $\mathcal{P} \in \mathbb{R}^{P \times 3}$ is initialized in uniform distribution:

$$z \sim \mathrm{U}(-r, +r), \tag{1}$$

where $z$ is the coordinates on the $z$-axis, and the other two dimensions are the same distribution. $r$ represents the range of a single dimension of coordinates.

Given a raw input point cloud $\boldsymbol{x} \in \mathbb{R}^{S \times 3}$ with $S$ points, firstly we hybrid Point Prompt $\mathcal{P} \in \mathbb{R}^{P \times 3}$ into its 3D space, denoted as $[\boldsymbol{x}; \mathcal{P}] \in \mathbb{R}^{(S+P) \times 3}$, where "[ ]" indicates concatenation and $P$ is the number of learnable points. Additionally, we modify the shape of $\boldsymbol{x}$ with our Point Shift Prompter, producing a shifted point cloud $\tilde{\boldsymbol{x}} \in \mathbb{R}^{S \times 3}$. This module also generates instance-specific informative shape features $\boldsymbol{f} \in \mathbb{R}^{D}$, where $D$ is the embedding dimension of transformers, formulated as:

$$\tilde{\boldsymbol{x}}, \boldsymbol{f} = \text{Point-Shift-Prompter}(\boldsymbol{x}). \tag{2}$$

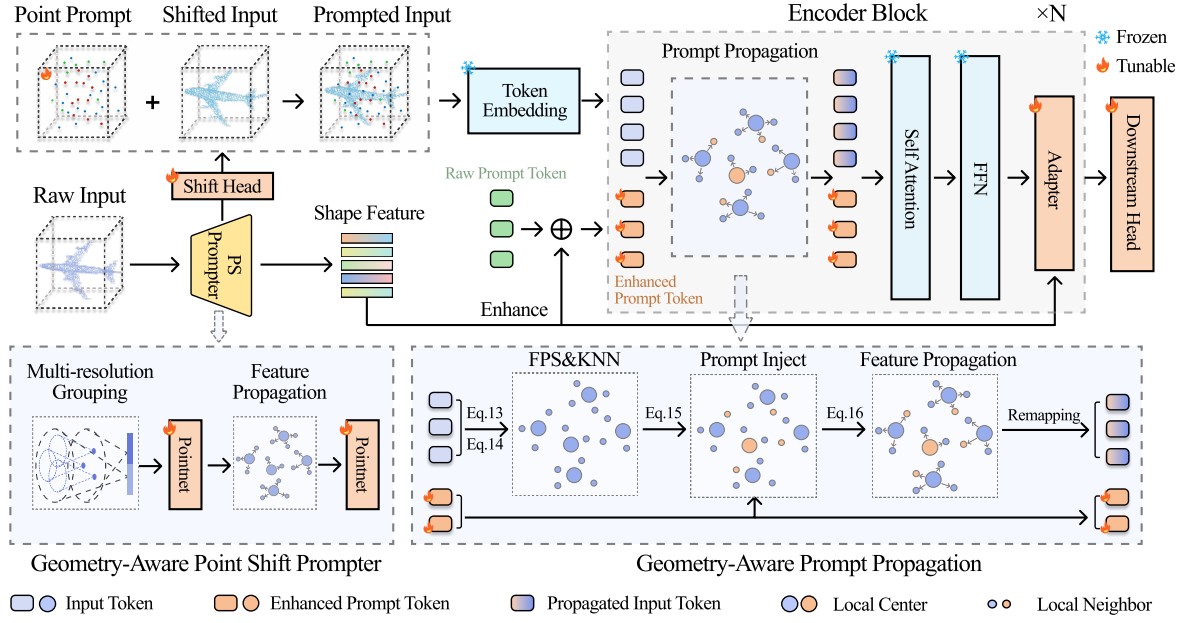

*Figure 3.* The overall pipeline of GAPrompt. The raw input point clouds are processed by Point Shift Prompter, generating instance-specific shape features and shifted points. These shifted points are then combined with the Point Prompt and embedded into input tokens. The shape features are further utilized to enhance the prompt tokens, which are concatenated with the input tokens. Subsequently, the concatenated tokens are fed into Prompt Propagation, incorporating the instance shape information into the model's feature extraction process. Finally, the propagated tokens are passed through pre-trained model blocks to produce results.

Then the hybrid point cloud $[\boldsymbol{x}; \boldsymbol{\mathcal{P}}]$ becomes prompted input point cloud $[\tilde{\boldsymbol{x}}; \boldsymbol{\mathcal{P}}] \in \mathbb{R}^{(S+P)\times 3}$.

Following the original architecture of the pre-trained model, the prompted point cloud is encoded into $L_t$ point tokens $\boldsymbol{h}_1$ by the token embedding module. We denote input tokens of $i$-th block as $\boldsymbol{h}_i \in \mathbb{R}^{L_t \times D}$. After that, input tokens $\boldsymbol{h}_i$ are concatenated with $L_p$ prompt tokens $\boldsymbol{p}_i \in \mathbb{R}^{L_p \times D}$ enhanced by shape feature $\boldsymbol{f}$, formulated as $[\boldsymbol{h}_i; \boldsymbol{p}_i]$. Then we feed these tokens into our Prompt Propagation mechanism, injecting prompt tokens into the feature extraction process:

$$\tilde{\boldsymbol{h}}_i = \text{Prompt-Propagation}([\boldsymbol{h}_i; \boldsymbol{p}_i]), \qquad (3)$$

where $\tilde{\boldsymbol{h}}_i \in \mathbb{R}^{L_t \times D}$ is the propagated input tokens. Finally, we feed $\tilde{\boldsymbol{h}}_i$ with enhanced prompt tokens $\boldsymbol{p}_i$ into vision transformer attention layers, consisting of self-attention and feed-forward layers. Furthermore, we adjust the tokens with adapters enhanced by shape feature $\boldsymbol{f}$.

$$\hat{\boldsymbol{h}}_i, \hat{\boldsymbol{p}}_i = \text{Attn.}([\tilde{\boldsymbol{h}}_i, \boldsymbol{p}_i]), \qquad (4)$$

$$\boldsymbol{h}_{i+1} = \hat{\boldsymbol{h}}_i + \text{Adapter}(\hat{\boldsymbol{h}}_i + \boldsymbol{f} \cdot \beta_a), \qquad (5)$$

where $\hat{\boldsymbol{h}}_i, \hat{\boldsymbol{p}}_i \in \mathbb{R}^{L_t \times D}$ are intermediate outputs of attention layer and $\beta_a$ is scale factor for enhancing adapters. In the following parts, we will demonstrate the details of Point Shift Prompter and Prompt Propagation respectively.

### 3.2. Point Shift Prompter

In addition to employing Point Prompt to capture subtle geometric details, we utilize a Point Shift Prompter to extract informative shape features from raw input point clouds. As described above, to adjust the point cloud shape accordingly, we pass point-level shape features through a shallow shift head to generate a coordinate shift for each input point. Furthermore, the shape features can be utilized to enhance the geometry awareness of prompt tokens and adapters.

**Shape Feature.** Specifically, to acquire global shape information of point clouds without much computational cost, we utilize a hierarchical downsampling strategy. As shown in Figure 3, the raw input point cloud $\boldsymbol{x}$ is sampled by multi-resolution grouping referring to PointNet++ (Qi et al., 2017b), which iteratively finds out $C_j$ center points $\boldsymbol{x}_j \in \mathbb{R}^{C_j \times 3}$ via farthest point sampling (FPS) at $j$-th resolution level. Then we respectively find out neighbor points $\boldsymbol{n}_j \in \mathbb{R}^{C_j \times K_j \times 3}$ corresponding to each center with $K$-nearest neighbor (KNN) algorithm:

$$\boldsymbol{x}_{j+1} = \text{FPS}(\boldsymbol{x}_j), \qquad (6)$$

$$\boldsymbol{n}_j = \text{KNN}(\boldsymbol{x}_j, \boldsymbol{x}_{j+1}). \qquad (7)$$

Obtaining such hierarchical spatial information, we embed point coordinates $\boldsymbol{x}_j$ to features $\boldsymbol{d}_j \in \mathbb{R}^{C_j \times D_j}$ with a lightweight pointnet (Qi et al., 2017a), which is a combina-

tion of convolutional layers, formulated as:

$$\tilde{\boldsymbol{d}}_j = \text{Pointnet}(\boldsymbol{x}_j). \qquad (8)$$

After $k$ levels of downsampling, we obtain center point features $\tilde{\boldsymbol{d}}_k \in \mathbb{R}^{C_k \times D_k}$ where $C_k \times D_k = D$ and concatenate them as shape feature $\boldsymbol{f}$,

$$\boldsymbol{f} = \text{Reshape}(\tilde{\boldsymbol{d}}_k) \in \mathbb{R}^D, \qquad (9)$$

which is the overall shape information of input point clouds.

**Shifted Input.** Furthermore, to generate instance-specific shifts for each point, we need to obtain features for all the original points $\boldsymbol{x}$. Firstly, an upsampling strategy is employed to propagate features from center points to neighbor points. Then we further process the features with another pointnet:

$$\tilde{\boldsymbol{d}}_j^n = \text{Pointnet}(\text{Propagate}(\tilde{\boldsymbol{d}}_j)), \qquad (10)$$

where $\tilde{\boldsymbol{d}}_j^n \in \mathbb{R}^{C_j \times K_j \times D_j}$ is features of neighbor points $n_j$ in $j$-th level. After upsampling from level $k$ to level $1$, the feature propagation process yields $\tilde{\boldsymbol{d}}_1^n$. We concatenate $\tilde{\boldsymbol{d}}_1^n$ and $\tilde{\boldsymbol{d}}_1$ and send into a small MLP, namely shift head, to generate the $\tilde{\boldsymbol{x}}$:

$$\tilde{\boldsymbol{x}} = \text{Shift-Head}([\tilde{\boldsymbol{d}}_1^n, \tilde{\boldsymbol{d}}_1]). \qquad (11)$$

Besides, the shape feature $\boldsymbol{f} \in \mathbb{R}^D$ can be used to enhance prompt tokens, imparting rich geometric awareness:

$$\boldsymbol{p}_i = \boldsymbol{p}_i' + \boldsymbol{f} \cdot \beta_p, \qquad (12)$$

where $\boldsymbol{p}_i' \in \mathbb{R}^{L_p \times D}$ denotes raw prompt tokens, $\boldsymbol{p}_i \in \mathbb{R}^{L_p \times D}$ is enhanced prompt tokens, $\beta_p$ is a scale factor.

### 3.3. Prompt Propagation

With prompt tokens enhanced by global shape features from the Point Shift Prompter, we further leverage these tokens to enhance the model's geometry awareness. To achieve this, we design a Prompt Propagation mechanism that integrates instance-specific shape information into the model's feature extraction process.

Given a set of input tokens $[\boldsymbol{h}_i; \boldsymbol{p}_i] \in \mathbb{R}^{(L_t + L_p) \times D}$ in the $i$-th block, we can find the geometric relationship between input tokens via FPS and KNN. We formulate it as follows:

$$\boldsymbol{h}_i^c = \text{FPS}(\boldsymbol{h}_i), \qquad (13)$$

$$\boldsymbol{h}_i^n = \text{KNN}(\boldsymbol{h}_i, \boldsymbol{h}_i^c). \qquad (14)$$

Subsequently, we randomly replace the enhanced prompt tokens $\boldsymbol{p}_i$ into input tokens in $\boldsymbol{h}_i^c$ and $\boldsymbol{h}_i^n$, namely prompt injection. $c$ and $n$ denote the index for center and neighbor points. Inspired by the thought of dropout (Srivastava et al.,

2014), the prompt injection process makes our model more robust while adapting downstream. A variety of replacement methods for alternation will be discussed in detail in the Ablation Study. We formulate this process as:

$$\boldsymbol{h}_i^{c'}, \boldsymbol{h}_i^{n'} = \text{Inject}(\boldsymbol{h}_i^c, \boldsymbol{h}_i^n, \boldsymbol{p}_i). \qquad (15)$$

where $\boldsymbol{h}_i^c \in \mathbb{R}^{C \times D}$, $\boldsymbol{h}_i^n \in \mathbb{R}^{C \times K \times D}$ denote tokens indexed by centers and neighbors and $\boldsymbol{h}_i^{c'}, \boldsymbol{h}_i^{n'}$ denote tokens after injection. Then we propagate features from local centers to all input tokens, formulated as:

$$\tilde{\boldsymbol{h}}_i = \text{Propagate}(\boldsymbol{h}_i^{c'}), \qquad (16)$$

where $\tilde{\boldsymbol{h}}_i$ is propagated input tokens. More details of Eq. 15&16 can be found in Appendix.

The core of our method lies in the strategic injection of enhanced prompt tokens. Without this enhancement, the propagation process among input tokens would result in trivial solutions, yielding minimal performance improvements. Specifically, when the features of center points are provided, and the goal is to compute the features of neighboring points, our propagation mechanism performs feature interpolation. This interpolation leverages the spatial distances between the center and neighboring points, allowing the model to effectively transfer and refine feature information across the tokens, thereby ensuring performance gains.

### 3.4. Analysis and Discussion

The objective of our method is to facilitate task-specific model adaptation through the integration of geometric-aware prompt mechanisms. In contrast to previous approaches, our method not only incorporates prompt tokens but also effectively captures fine-grained point-level geometric information. The attention mechanism with prompt integration can be formally expressed as follows:

$$\mathbf{o}_i = \text{Attn.}(W_Q \boldsymbol{h}_i, W_K \boldsymbol{h}_i, W_V \boldsymbol{h}_i), \qquad (17)$$

$$\hat{\mathbf{o}}_i = \text{Attn.}(W_Q \boldsymbol{h}_i, W_K [\boldsymbol{p}_i, \boldsymbol{h}_i], W_V [\boldsymbol{p}_i, \boldsymbol{h}_i]), \qquad (18)$$

where $\mathbf{o}_i$ and $\hat{\mathbf{o}}_i$ represent the attention outputs without and with prompt integration.

Building upon the theory established by (He et al., 2021), we can derive an equivalent transformation of Eq. 18 as:

$$\hat{\mathbf{o}}_i = \sum_{\mathbf{p}^k \in \boldsymbol{p}_i} A_{ik} W_V \mathbf{p}^k + (1 - \sum_k A_{ik}) \mathbf{o}_i, \qquad (19)$$

where $A_{ik}$ denotes the attention weight assigned to prompt $\mathbf{p}^k$ for query $\boldsymbol{h}_i$ by the transformer. This formulation reveals two critical aspects: the soft prompt mechanism induces a linear interpolation of the head's position-wise output, and the bias term facilitates adaptation within an offset subspace.

The key distinction of our approach lies in the point-level operation, addressing the limitations of previous prompting

*Table 1.* Classification on three variants of the ScanObjectNN and the ModelNet40, including the number of trainable parameters (Param) and overall accuracy (Acc). We report ScanObjectNN and ModelNet40 results without voting.

| Method | Reference | Param.(M) ↓ | FLOPs(G) ↓ | ScanObjectNN | | | ModelNet40 |
|---|---|---|---|---|---|---|---|
| | | | | OBJ_BG | OBJ_ONLY | PB_T50_RS | Acc. (%) ↑ |
| *Full Fine-Tuning* | | | | | | | |
| OcCo | ICCV 21 | 22.1 | 4.8 | 84.85 | 85.54 | 78.79 | 92.1 |
| Point-BERT | CVPR 22 | 22.1 | 4.8 | 87.43 | 88.12 | 83.07 | 92.7 |
| MaskPoint | ECCV 22 | 22.1 | - | 89.70 | 89.30 | 84.60 | 93.8 |
| Point-MAE | ECCV 22 | 22.1 | 4.8 | 90.02 | 88.29 | 85.18 | 93.2 |
| Point-M2AE | NeurIPS 22 | 15.3 | 3.6 | 91.22 | 88.81 | 86.43 | 93.4 |
| ReCon | ICML 23 | 43.6 | 5.3 | 94.15 | 93.12 | 89.73 | 93.9 |
| PointGPT-L | NeurIPS 23 | 360.5 | 67.7 | 97.20 | 96.60 | 93.40 | 94.1 |
| Point-FEMAE | AAAI 24 | 27.4 | 5.0 | 95.18 | 93.29 | 90.22 | 94.0 |
| *Efficient Fine-Tuning* | | | | | | | |
| Point-MAE | CVPR 22 | 22.1(100%) | 4.8 | 90.02 | 88.29 | 85.18 | 93.2 |
| +IDPT | ICCV 23 | 1.7(7.69%) | 7.2 | 91.22(+1.20) | 90.02(+1.73) | 84.94(-0.24) | 93.3(+0.1) |
| +DAPT | CVPR 24 | 1.1(4.97%) | 5.0 | 90.88(+0.86) | **90.19**(+1.90) | 85.08(-0.10) | 93.5(+0.3) |
| +Point-PEFT | AAAI 24 | 0.7(3.17%) | 7.0 | 89.33(-0.69) | 88.98(+0.69) | 84.42(-0.76) | **94.2**(+1.0) |
| **+GAPrompt** | **This Paper** | **0.6(2.71%)** | 5.0 | **91.91**(+1.89) | **90.19**(+1.90) | **85.57**(+0.39) | **94.2**(+1.0) |
| ReCon | ICML 23 | 43.6(100%) | 5.3 | 94.15 | 93.12 | 89.73 | 93.9 |
| +IDPT | ICCV 23 | 1.7(3.90%) | 7.2 | 93.29(-0.86) | 91.57(-1.55) | 87.27(-2.46) | 93.4(-0.5) |
| +DAPT | CVPR 24 | 1.1(2.52%) | 5.0 | 94.32(+0.17) | 92.43(-0.69) | 89.38(-0.35) | 93.5(-0.4) |
| +Point-PEFT | AAAI 24 | 0.7(1.61%) | 7.0 | 92.94(-1.21) | 91.57(-1.55) | 89.07(-0.66) | 93.8(-0.1) |
| **+GAPrompt** | **This Paper** | **0.6(1.38%)** | 5.0 | **94.49**(+0.34) | **92.60**(-0.52) | **89.76**(+0.03) | **94.0**(+0.1) |
| PointGPT-L | NeurIPS 23 | 360.5(100%) | 67.7 | 97.20 | 96.60 | 93.40 | 94.1 |
| +IDPT | ICCV 23 | 10.0(2.77%) | 75.2 | 98.11(+0.91) | 96.04(-0.56) | 92.99(-0.41) | 93.4(-0.7) |
| +DAPT | CVPR 24 | 4.2(1.17%) | 71.6 | 98.11(+0.91) | 96.21(-0.39) | 93.02(-0.38) | 94.2(+0.1) |
| +Point-PEFT | AAAI 24 | 3.1(0.86%) | 73.2 | 97.76(+0.56) | 96.21(-0.39) | 93.11(-0.29) | 93.9(-0.2) |
| **+GAPrompt** | **This Paper** | **2.0(0.55%)** | 71.8 | **98.97**(+1.77) | **96.73**(+0.13) | **94.31**(+0.91) | **96.2**(+2.1) |
| Point-FEMAE | AAAI 24 | 27.4(100%) | 5.0 | 95.18 | 93.29 | 90.22 | 94.0 |
| +IDPT | ICCV 23 | 1.7(6.20%) | 7.2 | 92.94(-2.24) | 90.88(-2.41) | 88.38(-1.84) | 93.4(-0.6) |
| +DAPT | CVPR 24 | 1.1(4.01%) | 5.0 | 93.98(-1.20) | 92.25(-1.04) | 88.51(-1.71) | 93.2(-0.8) |
| +Point-PEFT | AAAI 24 | 0.7(2.55%) | 7.0 | 94.32(-0.86) | 92.94(-0.35) | 89.35(-0.87) | 94.3(+0.3) |
| **+GAPrompt** | **This Paper** | **0.6(2.19%)** | 5.0 | **95.53**(+0.35) | **93.63**(+0.34) | **90.67**(+0.45) | **94.5**(+0.5) |

methods that primarily operate at the token level. While existing methods often struggle to capture fine-grained geometric features due to their token-level adaptation constraints, our Point Prompt and Point Shift Prompter mechanisms directly modulate $h_i$ in Eq. 17 and Eq. 18, enabling precise latent space adjustments at the point level. This fundamental difference in operational granularity contributes to enhanced geometric feature representation and more effective model adaptation for point cloud models.

## 4. Experiments

We evaluate the performance of our proposed GAPrompt on the point cloud classification task. We utilize four pretrained models Point-MAE (Pang et al., 2022), ReCon (Qi et al., 2023), Point-GPT (Chen et al., 2024) and Point-

FEMAE (Zha et al., 2024) as baselines. For a fair comparison, we employ identical data augmentation to the full fine-tuning method for each baseline. The hyperparameters are set as $\beta_a = 0.5$, $\beta_p = 0.5$ and $P = 20$. Additional analyses of $\beta_a$ and $P$ can be found in the Appendix.

### 4.1. Experimental Settings

**ScanObjectNN.** The ScanObjectNN (Uy et al., 2019) is a highly challenging 3D dataset comprising 15K real-world objects across 15 categories. These objects consist of indoor scene data obtained by scanning, exhibiting characteristics such as cluttered backgrounds and occlusions. As demonstrated in Table 1, we conducted experiments on three variants of ScanObjectNN, each with increasing complexity. Note that our experiments on dataset ScanObjectNN sample

*Table 2.* Comparisons of PEFT methods from NLP and 2D Vision on the hardest variant of ScanObjectNN.

| Method | Param. (M) | Acc. |
|---|---|---|
| Point-MAE | 22.1 | 85.18 |
| Linear Probing | 0.3 | 75.99 |
| Prefix Tuning | 0.7 | 77.72 |
| VPT | 0.4 | 81.09 |
| Adapter Tuning | 0.9 | 83.93 |
| LoRA | 0.9 | 81.74 |
| SSF | 0.4 | 82.58 |
| AdapterFormer | 0.9 | 83.45 |
| **GAPrompt** | 0.6 | **85.57** |

*Table 3.* The effect of components in our GAPrompt.

| Point Prompt | PS-Prompter | Prompt Propagation | Acc. |
|---|---|---|---|
| ✓ | - | - | 87.85 |
| ✓ | ✓ | - | 89.34 |
| ✓ | ✓ | ✓ | **90.67** |

*Table 4.* Ablation study on Point Shift Prompter.

| Shift Head | Prompt Enhance | Adapter Enhance | Acc. |
|---|---|---|---|
| ✓ | - | - | 88.23 |
| ✓ | ✓ | - | 89.71 |
| ✓ | ✓ | ✓ | **90.67** |

2048 points as input for each point cloud, consistent with previous works (Wang et al., 2021; Liu et al., 2022).

**ModelNet40.** ModelNet40 (Wu et al., 2015) comprises 12,311 pristine 3D CAD models across 40 categories, with complete, uniform, and noise-free point clouds that simplify the task. Following baselines, we sample 1024 points per instance. Since voting (Liu et al., 2019) is time-consuming, we focus on reporting overall accuracy without it.

## 4.2. Quantitative Analysis

**Performance on ScanObjectNN.** As shown in Table 1, our method GAPrompt achieves the highest accuracy among all the parameter-efficient fine-tuning methods for 3D vision models. Furthermore, we even surpass the full fine-tuning of Point-MAE, ReCon, Point-GPT, and Point-FEMAE by **1.89%, 0.34%, 1.77%, 0.35%** on OBJ_BG variant of ScanObjectNN respectively, and reduce over **97%** trainable parameters. It is attributed to our GAPrompt, which captures instance-specific geometry information of original point clouds and effectively integrates it into the feature extraction process of pre-trained models. In contrast, IDPT, DAPT, and Point-PEFT fall short of full fine-tuning performance due to their limited ability to capture geometric information from point clouds. Moreover, our method stands out in both efficiency and computational cost. With just 0.6M trainable parameters, our GAPrompt requires far fewer than IDPT and DAPT. In terms of FLOPs, our approach adds virtually no extra computational burden compared to baselines, significantly outperforming IDPT and Point-PEFT. This can be credited to our lightweight Point Shift Prompter and the parameter-free Prompt Propagation mechanism.

**Performance on ModelNet40.** As shown in Table 1, although the result in ModelNet40 is almost saturated, our GAPrompt still excels over previous works due to the instance-specific shape features extracted from point clouds and explicit awareness of point clouds. Moreover, GAPrompt gains positive increments on all four baselines

even with less than 3% trainable parameters and less than 0.1G FLOPs of computational increment, which verifies the efficacy and efficiency of adopting geometry information for prompting even on noiseless point clouds. Notably, our GAPrompt achieves the state-of-the-art performance of **96.2%** based on Point-GPT, with basic scale-and-translate augmentation and no voting.

**Comparison to PEFT Methods.** To further illustrate our superiority, we compare GAPrompt with several PEFT approaches (Chen et al., 2022; Lian et al., 2022) from NLP and 2D vision. As shown in Table 2, we select the most sophisticated PB_T50_RS variant with Point-MAE as the baseline. Although these basic methods show different improvements over linear probing, there is still a considerable performance gap to full fine-tuning due to the irregularity and complexity of point cloud structures. Our method with 3D-specific designs excels VPT and Adapter respectively by **4.48%** and **1.64%**, attributed to instance-specific shape features integration into the feature extraction of pre-trained models. Besides, our GAPrompt even has comparable trainable parameters against these basic methods, attributed to the compact design of Point Shift Prompter which directly extracts geometry features from point clouds.

## 4.3. Ablation Study

We conduct ablation studies on the most challenging PB_T50_RS variant based on Point-FEMAE to investigate the rationalization and effectiveness of our GAPrompt.

**Analysis on Main Components.** As shown in Table 3, to quantify the contribution of each main component, we incrementally conduct ablation experiments until the complete GAPrompt scheme. Solely introducing the Point Prompt, the performance can reach 87.85%, for facilitating explicit awareness of point clouds and assisting models in capturing subtle geometry. Then appending Point Shift Prompter leads to a 1.89% increment due to capturing instance-specific geometry information. Finally, the utilization of Prompt Propagation mechanism boosts the result to 90.67%, surpassing the full fine-tuning, which is attributed to the integrating enhanced prompts into feature extraction of pre-trained model.

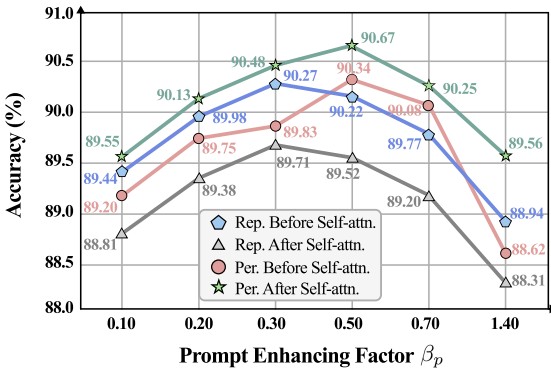

*Figure 4.* Ablation study of Prompt Propagation mechanism and prompt enhancing factor $\beta_p$.

**Effect of Point Shift Prompter Components.** As shown in Table 4, we evaluate the effectiveness of each component in our Point Shift Prompter in an incremental manner. Basically, we only use the Shift Head to produce shifted point clouds as input for the encoder, attaining 88.23% accuracy. Furthermore, respectively enhancing Prompt and Adapter with extracted shape features boosts the performance to the peak. This increment is attributed to the instance-specific shape features improving the geometry awareness of token prompts and adapters.

**Analysis on Settings of Prompt Propagation and $\beta_p$.** As shown in Figure 4, we conduct ablation experiments on prompt propagation settings and prompt enhancing factor $\beta_p$. Our prompt propagation mechanism has two alternations, operating before or after self-attention layers, denoted as 'Before Self-attn.' and 'After Self-attn.'. Furthermore, we design two different prompt injection options, which are realized by directly replacing and indirectly permutation. The combination of these choices produces the four curves in Figure 4. It is clear that utilizing permutation after self-attention layers and 0.5 as $\beta_p$ obtains the peak performance. Intuitively, it is because this setting brings more randomness and results in more robust convergence.

**Visualization of Attention Position.** In Figure 5, we respectively visualize the attention positions of the [CLS] token of GAPrompt and full fine-tuning, where the warm color indicates higher values. As illustrated, the [CLS] token of GAPrompt captures more essential 3D semantics, such as the head and vertical stabilizer of the plane, the stand of the lamp, and the pot of the flower pot. But the [CLS] token of full fine-tuning merely grasps the vertical tails and the base of the lamp. It indicates that with inherent geometric clues in point clouds, GAPrompt effectively grasps the critical information and further benefits point cloud understanding.

**Visualization of Shifted Point Clouds.** Figure 6 depicts the raw and shifted point clouds from the test split of ScanObjectNN. The raw point clouds are noisy and scattered, reflecting the inherent complexity of real-world data. Note the green color mapping is merely for convenient recognition.

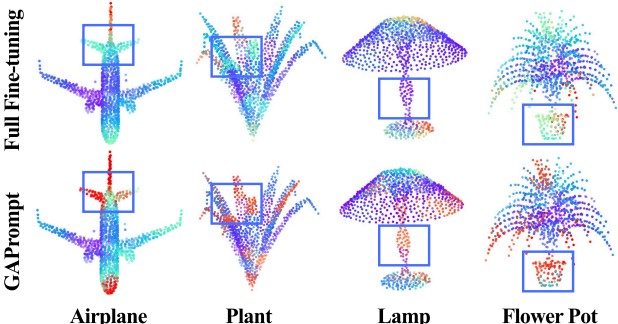

*Figure 5.* Visualization of attention position of GAPrompt and full fine-tuning. We visualize the attention scores of the [CLS] token to other point cloud tokens.

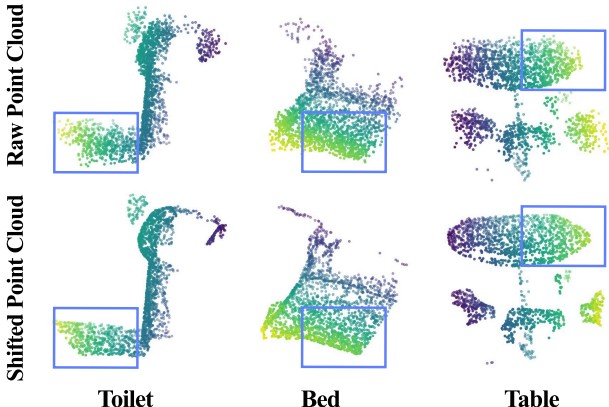

*Figure 6.* Visualization of point clouds before and after transformation by the Point Shift Prompter. The samples are drawn from the test split of the ScanObjectNN dataset, demonstrating its broad generalization capability across unseen data.

We can see that point clouds shifted by Point Shift Prompter tend to be more compact and exhibit sharper boundaries, making them easier to recognize, especially in the regions highlighted by the rectangle. This suggests that the Point Shift Prompter can enhance the geometric features of the point cloud at the input level, thereby contributing to improved performance. Additionally, the point cloud contains learnable point prompts, which tend to move to the inner space of point clouds during training.

## 5. Conclusion

In this paper, we introduce Geometry-Aware Point Cloud Prompt (GAPrompt), parameter-efficient fine-tuning specific to pre-trained 3D vision models. We find that capturing instance-specific shape features is an effective way to enhance geometry awareness of prompting. To capture and utilize inherent geometric clues in point clouds, we readily develop the Point Prompt, the Point Shift Prompter, and the Prompt Propagation mechanism, greatly boosting the representational ability of prompting. Our approach outperforms other state-of-the-art parameter-efficient fine-tuning methods and reduces trainable parameters significantly.

## Acknowledgments

This work was supported by the National Natural Science Foundation of China (62376011) and the National Key R&D Program of China (2024YFA1410000).

## Impact Statement

This paper presents work whose goal is to advance the field of Machine Learning. There are many potential societal consequences of our work, none of which we feel must be specifically highlighted here.

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

# Appendix

## A. Training Detail

We adopt downstream fine-tuning configurations in alignment with the pioneering work Point-MAE (Pang et al., 2022). The detailed configurations are provided in Table 5. For example, when fine-tuning on ScanObjectNN (Uy et al., 2019), the training process spans 400 epochs, using a cosine learning rate scheduler (Loshchilov & Hutter, 2022) that starts at 5e-4, with a 10-epoch warm-up period. The AdamW optimizer (Loshchilov & Hutter, 2019) is employed. Given that ReCon (Qi et al., 2023) and Point-FEMAE (Zha et al., 2024) extend Point-MAE with several additional modules, we follow the approach of DAPT (Zhou et al., 2024) by only loading pre-trained weights into a Point-MAE model for efficient fine-tuning, while excluding the residual components of ReCon and Point-FEMAE. This option also leads to a sight computational saving, as shown in Table 1. Indeed, the computation overhead should be calculated by subtracting FLOPs of fine-tuning Point-MAE. All experiments are conducted on a single GeForce RTX 4090 using PyTorch version 1.13.1.

*Table 5.* Training details for downstream fine-tuning.

| Dataset | ScanObjectNN | | | ModelNet |
|---|---|---|---|---|
| | OBJ_BG | OBJ_ONLY | PB_T50_RS | 1K |
| Optimizer | AdamW | AdamW | AdamW | AdamW |
| Learning rate | 5e-4 | 5e-4 | 5e-4 | 5e-4 |
| Weight decay | 5e-2 | 5e-2 | 5e-2 | 5e-2 |
| Learning rate scheduler | cosine | cosine | cosine | cosine |
| Training epochs | 400 | 400 | 400 | 400 |
| Warmup epochs | 10 | 10 | 10 | 10 |
| Batch size | 32 | 32 | 32 | 32 |
| Point Prompt number | 20 | 10 | 20 | 5 |
| Prompt enhancing factor | 0.5 | 0.5 | 0.5 | 0.5 |
| Adapter enhancing factor | 0.5 | 0.5 | 0.5 | 0.5 |
| Number of points | 2048 | 2048 | 2048 | 1024 |
| Number of point patches | 128 | 128 | 128 | 64 |
| Point patch size | 32 | 32 | 32 | 32 |

## B. Implementation Detail

### B.1. Prompt Injection

We provide a comprehensive elucidation of the Prompt Injection process, as delineated in Eq. 15 of the main paper.

Given a set of input tokens and prompt tokens $[\boldsymbol{h}_i; \boldsymbol{p}_i] \in \mathbb{R}^{(L_t+L_p) \times D}$ in the $i$-th block, we can find geometric relationship between input tokens via FPS and KNN algorithm. We use $\boldsymbol{h}_i^c \in \mathbb{R}^{C \times D}$ and $\boldsymbol{h}_i^n \in \mathbb{R}^{C \times K \times D}$ to denote center tokens and local neighboring tokens, where $c$ and $n$ represent the index for center and neighboring tokens. $L_t$ is the input tokens number, $L_p$ is the prompt tokens number, and $C$ and $K$ represent the number of local centers and local neighbors respectively.

Inspired by the thought of dropout (Srivastava et al., 2014), the prompt injection process randomly replaces the center and neighboring tokens with enhanced prompt tokens, which are informative of global shape.

Considering that FPS randomly selects initial points, although the resulting sets are consistent, the permutation of the resulting point sets may vary due to inherent randomness. Leveraging this randomness, we replace the last $L_p$ tokens in $\boldsymbol{h}_i^c$ with $\boldsymbol{p}_i$, and similarly, replace the last $L_p$ tokens in $\boldsymbol{h}_i^n$ with $\boldsymbol{p}_i$, injecting prompt tokens into input tokens. This prompt injection configuration is referred to as 'Replacement'.

Alternatively, another prompt injection method, termed 'Permutation', can be employed. Given that $\boldsymbol{h}_i^c$ and $\boldsymbol{h}_i^n$ are subsets of $\boldsymbol{h}_i$ indexed by $c$ and $n$, we can insert $L_p$ prompt tokens $\boldsymbol{p}_i$ before $\boldsymbol{h}_i$ and remove the last $L_p$ tokens of $\boldsymbol{h}_i$. Subsequently, by indexing the mixed set of $\boldsymbol{p}_i$ and $\boldsymbol{h}_i$ using $c$ and $n$, we obtain the tokens $\boldsymbol{h}_i^c$ and $\boldsymbol{h}_i^n$ with prompt tokens injected.

## B.2. Feature Propagation

In this paper, we extensively employ the Feature Propagation operation introduced in PointNet++ (Qi et al., 2017b). Below, we provide a detailed formulation.

Given the features of the center points, the objective is to compute the features of neighboring points through a propagation mechanism that performs feature interpolation. This interpolation utilizes the spatial distances between the center and neighboring points, enabling the model to efficiently transfer and refine feature information across the tokens.

We denote the set of center point coordinates as $c_i \in \mathbb{R}^3$, where $i = 1, \ldots, C$, and the coordinates of any neighboring point as $x \in \mathbb{R}^3$, where $C$ represents the number of center points. The corresponding feature of a given point is denoted as $f(\cdot)$. Our objective is to compute $f(x)$ based on $x$, $c_i$, and $f(c_i)$.

First, we compute the Euclidean distance from $x$ to each center point $c_i$:

$$d(x, c_i) = \|x - c_i\|. \tag{20}$$

Next, we calculate the weight by taking the inverse of the spatial distance:

$$w(x, c_i) = \frac{1}{d(x, c_i)^p}, \tag{21}$$

where $p$ is typically set to 2. This results in a set of weights $w(x, c_i)$ for $i = 1, \ldots, C$. We then select only the top-$K$ weights for interpolation:

$$\{w(x, c_j)\} = \text{Top-}K(\{w(x, c_i)\}), \tag{22}$$

where $j = 1, \ldots, K$, and $K$ is typically set to 32. Subsequently, the interpolation of features is based on the weighted distances, formulated as:

$$f(x) = \frac{\sum_{j=1}^{K} w(x, c_j) f(c_j)}{\sum_{j=1}^{K} w(x, c_j)}. \tag{23}$$

Finally, this procedure is repeated for each neighboring point to obtain their features for further utilization.

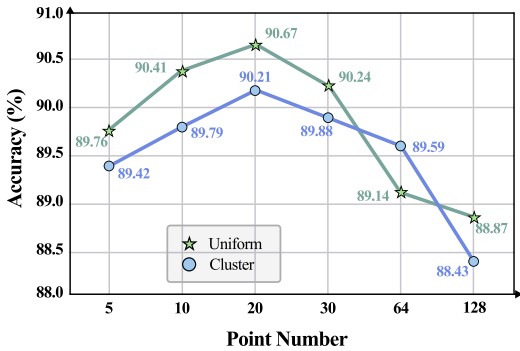
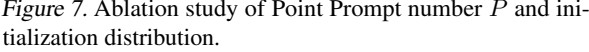

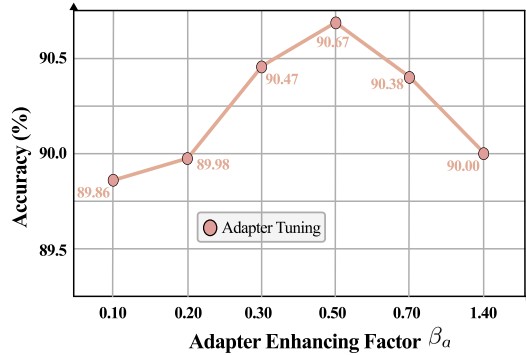

*Figure 7.* Ablation study of Point Prompt number $P$ and initialization distribution.

*Figure 8.* Ablation study on the impact of the adapter enhancing factor $\beta_a$ when augmented with shape features.

## C. Additional Experiments

### C.1. Analysis on Point Prompt Number $P$ and Initialization.

We perform quantitative analysis on Point Prompt about two additional different initialization distributions and learnable point number $P$. We respectively test initializing all points in a uniform distribution and in cluster forms, where each cluster follows a Gaussian distribution in 3D space. And we explore the impact of different point numbers. As shown in Figure 7, The result turns out that uniform distribution generally outperforms the cluster distribution, and with 20 learnable points, the model achieves the best results. This suggests that appropriate additional point prompts help to better capture subtle geometric features and to acquire explicit awareness of geometry.

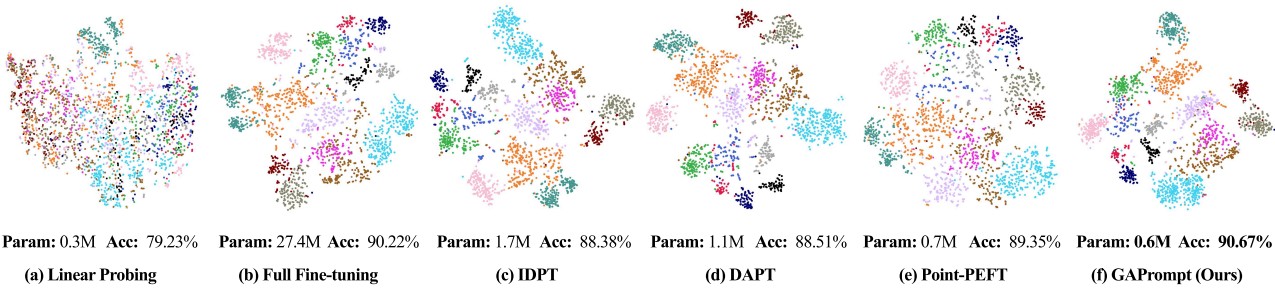

Param: 0.3M  Acc: 79.23%    Param: 27.4M Acc: 90.22%    Param: 1.7M Acc: 88.38%    Param: 1.1M Acc: 88.51%    Param: 0.7M  Acc: 89.35%    Param: 0.6M Acc: **90.67%**

(a) Linear Probing     (b) Full Fine-tuning     (c) IDPT     (d) DAPT     (e) Point-PEFT     (f) GAPrompt (Ours)

*Figure 9.* The t-SNE visualizations from the test sets of ScanObjectNN (PB_T50_RS) using a pre-trained Point-FEMAE with different tuning strategies. We extract the final classification features from the top linear layer for t-SNE visualizations.

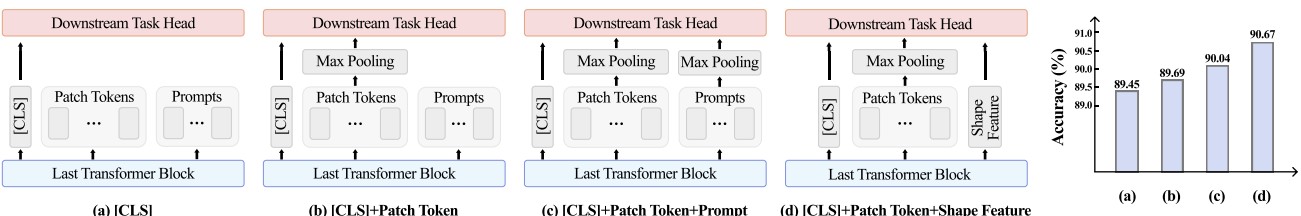

*Figure 10.* Ablation study on different input for downstream head.

## C.2. Analysis on Adapter Enhancing Factor $\beta_a$.

As shown in Figure 8, we conduct further ablation study on $\beta_a$, which is the factor when enhancing adapters with global shape features. It turns out that $\beta_a$ set to 0.5 leads to peak performance. This indicates that properly enhancing adapters with global shape features can boost the model's performance. We attribute it to that global shape features help adjust and refine feature distributions and improve geometry awareness.

## C.3. t-SNE Visualization of [CLS] Feature.

In Figure 9, the t-SNE (Van der Maaten & Hinton, 2008) feature manifold visualization displays the methods following linear probing, full fine-tuning, IDPT, DAPT, Point-PEFT and our GAPrompt on the ScanObjectNN PB_T50_RS dataset. From Figure 9 (a), it is evident that the feature distribution extracted by the pre-trained model appears less discriminative. We contend that this is mainly due to the significant domain gap between the synthetic pre-training ShapeNet dataset and the real-world ScanObjectNN dataset, demonstrating the necessity for adapting downstream tasks. With full fine-tuning in Figure 9 (b), the feature distribution becomes more discriminative as all parameters are tuned. Figure 9 (c-f) confirms that our GAPrompt helps the pre-trained model generate more distinguishable representations, with fewer learnable parameters and higher accuracy than other PEFT methods.

## C.4. Ablation on Downstream Head Input.

As demonstrated in Figure 10, we examine the impact of different input choices for the downstream head. We consider four options: the [CLS] token, the maximum input patch token, the maximum prompt token, and the global shape feature extracted by our Point Shift Prompter. Empirically, we find that the combination of the [CLS] token, the maximum patch token, and the shape feature yields the best results, demonstrated in Figure 10 (d). This can be attributed to the fact that the [CLS] token and the maximum patch token provide valuable knowledge extracted from the pre-trained model, while the shape feature introduces instance-specific global geometry information. However, in Figure 10 (c), the knowledge embedded in the maximum prompt token largely overlaps with that of the maximum patch token. Consequently, our shape feature complements the traditional outputs from the pre-trained model effectively, thereby enhancing the performance of the downstream head.

