# OpenReview forum: "GAPrompt: Geometry-Aware Point Cloud Prompt for 3D Vision Model"
_ICML.cc/2025/Conference — ICML 2025 poster_

### Official Review · Reviewer_7QpT · 2025-03-11

**Overall Recommendation:** 3

**Summary:**

The paper focuses on parameter-efficient fine-tuning for 3D vision models and introduces a geometry-aware prompt learning method, named GAPrompt. GAPrompt incorporates three key designs to effectively capture geometric information, including Point Prompt, Point Shift Prompter, and the Prompt Propagation mechanism. Experimental results on the ScanObjectNN and ModelNet40 datasets demonstrate the effectiveness of GAPrompt.

**Claims And Evidence:**

This paper emphasizes the importance of geometric information in parameter-efficient fine-tuning of 3D vision models. Experimental results demonstrate the effectiveness of the three proposed geometry-aware designs on benchmark datasets.

**Essential References Not Discussed:**

The Positional Prompt Tuning [PPT'22] method is designed for the same task as this work and has released models and code. However, it is neither cited nor compared in the paper.

[PPT'22] Positional Prompt Tuning for Efficient 3D Representation Learning.

**Experimental Designs Or Analyses:**

I have reviewed all the experiments.

Ablation studies in Tables 3 and 4 assess the effectiveness of each design. However, the effectiveness of Point Prompt is not explicitly verified in Table 3.

**Methods And Evaluation Criteria:**

Yes. The three proposed designs focus on capturing the geometric information, and ScanObjectNN and ModelNet40 are standard benchmark datasets to evaluate fine-tuning on 3D vision models.

**Other Comments Or Suggestions:**

None.

**Other Strengths And Weaknesses:**

### Strengths
- GAPrompt introduces a geometry-aware point cloud prompt for parameter-efficient fine-tuning of 3D vision models, incorporating three designs to effectively capture geometric information.
- Experimental comparisons with existing methods, along with ablation studies on individual designs, demonstrate the effectiveness of GAPrompt.
- The paper is well-written and easy to follow.

### Weaknesses
- Insufficient ablation study on Point Prompt in Table 3.
- Missing comparisons with [PPT'22] (Positional Prompt Tuning for Efficient 3D Representation Learning).
- Lack of few-shot learning experiments on ModelNet40 and part segmentation experiments on ShapeNetPart.

**Questions For Authors:**

### Major
- 1. The working mechanism of Point Prompt in capturing subtle geometric features is not clearly explained. In Table 3, results for "GAPrompt without Point Prompt" should be included. Additionally, besides analyzing the effect of different numbers of Point Prompts, visualizations of the trained Point Prompt should be provided for better understanding.

- 2. A comparison with [PPT'22] (Positional Prompt Tuning for Efficient 3D Representation Learning) should be added to Table 1.

- 3. It is recommended to include few-shot learning experiments on ModelNet40 and part segmentation experiments on ShapeNetPart.

### Minor

- 4. In Fig. 4, the results before and after Self-Attention are inconsistent across different prompt injection options. Further clarification is needed.

- 5. The number of Point Prompts varies across datasets in Table 5. A justification for this choice is needed. Additionally, details on how $L_p$ is set and how $p_i^{\prime}$ is initialized should be provided.

My evaluation is primarily based on the main weaknesses outlined in the previous section, which correspond to the major problems. Addressing these concerns with substantial evidence and clarifications could potentially influence my rating.

**Relation To Broader Scientific Literature:**

Prior studies have also acknowledged the importance of 3D-aware parameter-efficient fine-tuning. For example, IDPT'23 generates dynamic prompt tokens for each point cloud instance to capture semantic prior features, while DAPT'24 combines dynamic scale adapters with internal prompts for effective point cloud transfer learning.

**Theoretical Claims:**

Yes, I have checked the theoretical analysis in Sec. 3.4.

---

> ### Author Rebuttal · Authors · 2025-03-31
>
> ### Q1. working mechanism of Point Prompt
> As for working mechanism of Point Prompt, it can be analyzed through Equation 18 and 19 in paper. Previous prompting methods operate at **token level**, which corresponds to local patches, failing to adjust exact points within patches.
>
> In contrast, our Point Prompt directly prompts within patches, operating at **point-level grains**. When adapting to downstream, it is optimized via task loss to focus on critical points that encode subtle yet essential geometric information.
>
> As for visualization, the learned Point Prompts are already present in Figures 5 and 6. However, as there are only **20 prompts** within **2048 points**, may difficult to see. In final version, we will explicitly highlight learned Point Prompts in visualization for better clarity.
>
> ### W1. ablation study on Point Prompt in Table 3
> We supplement additional ablation results on "GAPrompt without Point Prompt" as shown below. Incorporating the learnable Point Prompt yields an **1.02%** gain in accuracy.
>
> |Point Prompt|Point Shift Prompter|Prompt Propagation|Acc.(%)|
> |:----------:|:------------------:|:----------------:|:------:|
> |-|-|-|86.10|
> |√|-|-|87.85|
> |√|√|-|89.34|
> |-|√|√|89.65|
> |√|√|√|90.67|
>
> ### Q2,W2. comparison with PPT
> Thanks for your advice, and we will add it to main table in final version. The primary reason we did not initially include this comparison is that PPT hasn't been officially published and only available on **arXiv**. Even though, we compare on 4 different representative backbones across 4 datasets, and our GAPrompt achieves **14** higher performances in 16 experiments with fewer trainable parameters(**0.6M**), due to our lightweight geometry-aware prompting design.
>
> |Method|Ref.|Param.|OBJ_BG|OBJ_ONLY|PB_T50_RS|ModelNet|
> |:-------:|:--------:|:-----:|:-----------:|:-------:|:-------:|:------:|
> ||||*Point-MAE*||||
> |+PPT|arXiv|1.1|89.33|88.81|84.87|93.7|
> |+GAPrompt|ThisPaper|**0.6**|**91.91**|**90.19**|**85.57**|**94.2**|
> ||||*ReCon*||||
> |+PPT|arXiv|1.1|**95.01**|**93.28**|89.52|93.8|
> |+GAPrompt|ThisPaper|**0.6**|94.49|92.60|**89.76**|**94.0**|
> ||||*PointGPT-L*||||
> |+PPT|arXiv|3.6|98.28|96.21|94.10|95.1|
> |+GAPrompt|ThisPaper|**2.0**|**98.97**|**96.73**|**94.31**|**96.2**|
> ||||*Point-FEMAE*||||
> |+PPT|arXiv|1.1|93.98|92.08|88.79|93.3|
> |+GAPrompt|ThisPaper|**0.6**|**95.53**|**93.63**|**90.67**|**94.5**|
>
> ### W3,Q3. few-shot and part segmentation experiments
> We supplement few-shot experiments on ModelNet40 and segmentation experiments on ShapeNetPart below.
> We compare our method with other SOTA works based on pre-trained Point-FEMAE. It can be seen that our method performs well in few-shot setting and can generalize to segmentation tasks well, verifying the efficacy of GAPrompt.
>
> *Few-shot on ModelNet40*
>
> ||Ref.|5-way||10-way||
> |:---------:|:--------:|:----------:|:----------:|:----------:|:----------:|
> |||10-shot|20-shot|10-shot|20-shot|
> |+DAPT|CVPR24|96.6±2.1|97.9±2.7|92.1±3.4|94.9±3.3|
> |+Point-PEFT|AAAI24|96.8±2.6|98.1±2.5|92.4±3.2|95.0±.3.1|
> |+PPT|arXiv|96.9±1.9|98.0±2.9|91.9±3.6|95.2±3.0|
> |+GAPrompt|ThisPaper|**97.2**±1.7|**98.4**±2.1|**92.7**±2.9|**95.7**±2.8|
>
> Result in *ShapeNetPart*
>
> |Method|Ref.|Param.|Cls. mIoU|Ins. mIoU|
> |:-------:|:--------:|:---------:|:-------:|:-------:|
> |||*Point-MAE*|||
> |+DAPT|CVPR24|5.65|84.01|85.7|
> |+PPT|arXiv|5.62|84.07|85.7|
> |+GAPrompt|ThisPaper|**5.55**|**84.10**|**85.8**|
> |||*ReCon*|||
> |+DAPT|CVPR24|5.65|83.87|85.7|
> |+PPT|arXiv|5.62|**84.23**|85.6|
> |+GAPrompt|ThisPaper|**5.55**|83.90|**85.8**|
>
> ### Q4. difference before and after Self-Attn in Fig. 4
> The difference before and after Self-Attention **arises from distinct inference orders**. This is similar to choosing between **"Attn→FFN"** or **"FFN→Attn"** in Transformer models. The Prompt Propagation mechanism enhances spatial information, further facilitating interactions between prompts and point tokens. When applied **after Self-Attention**, it provides stronger performance improvements.
>
> ### Q5a.  determination of Point Prompt number
> The number of Point Prompts is set **proportionally** to the input point cloud resolution, typically **0.5%-1%** of the dataset resolution.
> For **clean datasets** (e.g., ModelNet40, OBJ_ONLY), **0.5%** yields slightly better results (+0.1%), corresponding to 5 and 10 prompts for 1024 and 2048 points, respectively. While for **noisier datasets** (e.g., OBJ_BG, PB_RS_T50), **1%** provides a small gain (+0.2%), resulting in 20 prompts for 2048 points.
>
> ### Q5b. setting of $L_p$ and $p_i^{\prime}$
>  $L_p$ is set to **10**, as shown in the ablation study below.
> When $L_p$ is too small, the prompting effect is insufficient, leading to suboptimal performance. Conversely, an excessive $L_p$ does not yield further improvements while incurring additional parameter overhead. As for $p_i^{\prime}$, we use Kaiming Uniform initialization.
>
> |$L_p$|0|3|6|10|15|20|
> |-----|:---:|:---:|:---:|:-------:|:---:|:---:|
> |Acc.|89.34|89.98|90.34|**90.67**|90.55|90.47|

---

> > ### Comment · Reviewer_7QpT · 2025-04-09
> >
> > Thank you to the authors for their responses, which I have read carefully. In their responses, they added comparisons with PPT. However, some PPT results are inconsistent with those in their paper for the classification task with PointMAE and the few-shot setting with RECON, yet no explanation has been provided. This inconsistency is quite confusing and raises concerns for me,  leading me to lean toward a borderline rejection of the paper.

---

> > > ### Author Response · Authors · 2025-04-09
> > >
> > > We sincerely thank Reviewer **7QpT** for the additional feedback. We are sorry to see a downgrade from *weak accept* to *weak reject*, and we would like to offer further clarification regarding the two results in question.
> > >
> > > **Regarding the classification task of PPT on Point-MAE:**
> > >
> > > The discrepancy arises from the use of **different data augmentation strategies**. Specifically, the PPT [arXiv] paper employs **rotation augmentation**, which is relatively strong. In contrast, our paper and rebuttal adopt the same **scale and translation augmentation** used by Point-MAE and other PEFT methods to ensure fair and consistent comparisons across all baselines.
> > >
> > > To directly address your concern, we re-evaluated PPT under the **same augmentation setting (scale and translate)** using the official codebase, and reported the updated results in our rebuttal.
> > >
> > > We would also like to note that PPT is currently an **unpublished preprint on arXiv**, and thus, we were **not obliged to include it as a baseline** in our main paper. Nevertheless, we still conducted additional comparisons in good faith, aiming to meet the expectations raised in your review. We appreciate your attention to experimental rigor.
> > >
> > > **Regarding the few-shot task:**
> > >
> > > The confusion here stems from the use of **different backbones**. As stated in our rebuttal, the few-shot results we reported are based on the more recent and stronger backbone **Point-FEMAE [AAAI 2024]**, **not ReCon [ICML 2023]**. This was clarified in our response as: "*We compare our method with other SOTA works based on pre-trained Point-FEMAE.*"
> > >
> > > We hope these clarifications have addressed your concerns. Please kindly let us know if there are any remaining issues or if further clarification is needed. We value your feedback and are committed to improving the clarity and reproducibility of our work.

---

### Official Review · Reviewer_LHwj · 2025-03-13

**Overall Recommendation:** 4

**Summary:**

This is a paper on efficient point cloud fine-tuning, where the author added a geometric perception structure to the point cloud embedding part for efficient fine-tuning and achieved good results.

**Claims And Evidence:**

Yes

**Essential References Not Discussed:**

Some papers have not been compared and cited. The author should add these discussions, which will not affect the contribution of this paper.

Parameter-Efficient Fine-Tuning in Spectral Domain for Point Cloud Learning
Positional Prompt Tuning for Efficient 3D Representation Learning

**Experimental Designs Or Analyses:**

Yes

**Methods And Evaluation Criteria:**

Yes

**Other Comments Or Suggestions:**

None

**Other Strengths And Weaknesses:**

This work is solid and worth accepting, which has many ablation studies and visualization.

**Questions For Authors:**

1. How about the inference time and flops compared to the other works?
2. How many tokens of the Enhanced Prompt Token are concatenated to the original tokens in sequence? Is there any ablation study?

**Relation To Broader Scientific Literature:**

Previous point cloud PEFT methods, such as DAPT, IDPT, and PPT, mainly rely on prompt learning and adapter for feature extraction.

**Theoretical Claims:**

Yes, but this paper does not contain complex proofs.

---

> ### Author Rebuttal · Authors · 2025-03-31
>
> Thanks for your appreciation and valuable advice!
>
> ### Q1. inference time and FLOPs comparison
>
> We test the inference time and FLOPs of our method and other SOTA methods, including IDPT[ICCV23], DAPT[CVPR24] and Point-PEFT[AAAI24]. All experiments are conducted on a RTX 4090.
>
> As table shown below, our method attains the fastest inference time with only **11.1** millisecond per sample and fewest FLOPs with **5.0G** MACs among parameter-efficient works. Notably, our method merely introduces **4%** computational overhead but brings over **97%** reduction in trainable parameters.
>
> |                            | *Point-MAE* | +IDPT | +DAPT | +Point-PEFT | +GAPrompt |
> | :------------------------: | :---------: | :---: | :---: | :---------: | :-------: |
> | inference time (ms/sample) |    10.2     | 16.2  | 12.0  |    13.5     | **11.1**  |
> |         FLOPs (G)          |     4.8     |  7.2  |  5.0  |     7.0     |  **5.0**  |
>
> ### Q2. number of the enhanced prompt tokens
>
> The number of enhanced prompt tokens in our experiments is set to **10** as a hyperparameter. We provide additional ablation experiment results below and will add it into final version for better interpretability.
>
> We find that our method produces peak performance when adopting 10 enhanced prompt tokens. A smaller number leads to suboptimal prompting effects due to insufficient guidance, while an excessive number does not yield further improvements but incurring additional computational cost.
>
> | Prompt Token Number |   0   |   3   |   6   |    10     |  15   |  20   |  30   |
> | ------------------- | :---: | :---: | :---: | :-------: | :---: | :---: | :---: |
> | Acc. on PB_T50_RS   | 89.34 | 89.98 | 90.34 | **90.67** | 90.55 | 90.47 | 90.33 |
>
> ### C1. citation to PointGST and PPT
>
> Thanks for your reminder and we will cite them in final version. Notably, at the time of our submission and even at present, both works remain as preprints on arXiv and have not been officially published. However, our method still excels both in terms of accuracy and efficiency.
>
> We compare with them on four dataset based on four representative backbones, as shown below. GAPrompt achieve **12** SOTA results while PPT and PointGST get 2 SOTA results respectively. And our method has only **0.6M** trainable parameters, attaining highest parameter efficiency, attributed to our geometry-aware point-level prompting design.
>
> |  Method   |    Ref.    | Param.  |    OBJ_BG     | OBJ_ONLY  | PB_T50_RS | ModelNet |
> | :-------: | :--------: | :-----: | :-----------: | :-------: | :-------: | :------: |
> |           |            |         |  *Point-MAE*  |           |           |          |
> |   +PPT    |   arXiv    |   1.1   |     89.33     |   88.81   |   84.87   |   93.7   |
> | +PointGST |   arXiv    |   0.6   |     91.74     |   90.19   |   85.29   |   93.5   |
> | +GAPrompt | This Paper | **0.6** |   **91.91**   | **90.19** | **85.57** | **94.2** |
> |           |            |         |    *ReCon*    |           |           |          |
> |   +PPT    |   arXiv    |   1.1   |   **95.01**   | **93.28** |   89.52   |   93.8   |
> | +PointGST |   arXiv    |   0.6   |     94.49     |   92.94   |   89.49   |   93.6   |
> | +GAPrompt | This Paper | **0.6** |     94.49     |   92.60   | **89.76** | **94.0** |
> |           |            |         | *PointGPT-L*  |           |           |          |
> |   +PPT    |   arXiv    |   3.6   |     98.28     |   96.21   |   94.10   |   95.1   |
> | +PointGST |   arXiv    |   2.4   |     98.97     | **97.59** | **94.83** |   94.8   |
> | +GAPrompt | This Paper | **2.0** |   **98.97**   |   96.73   |   94.31   | **96.2** |
> |           |            |         | *Point-FEMAE* |           |           |          |
> |   +PPT    |   arXiv    |   1.1   |     93.98     |   92.08   |   88.79   |   93.3   |
> | +PointGST |   arXiv    |   0.6   |     94.66     |   92.94   |   90.22   |   93.8   |
> | +GAPrompt | This Paper | **0.6** |   **95.53**   | **93.63** | **90.67** | **94.5** |

---

### Official Review · Reviewer_WVi3 · 2025-03-13

**Overall Recommendation:** 3

**Summary:**

This paper proposes GAPrompt, a geometry-aware prompt learning method for parameter-efficient fine-tuning (PEFT) of pre-trained 3D vision models. Existing PEFT approaches in 3D vision struggle to capture geometric information from sparse and irregular point clouds. To address this, GAPrompt introduces three key innovations:

Point Prompt: Explicitly incorporates learnable point clouds as auxiliary input to enhance geometric awareness.

Point Shift Prompter: Dynamically adjusts point cloud positions using global shape features extracted via hierarchical downsampling.

Prompt Propagation: Integrates shape information into feature extraction through token replacement and interpolation.
Experiments on ScanObjectNN and ModelNet40 show GAPrompt achieves competitive performance with full fine-tuning (e.g., 90.67% vs. 90.22% on PB-T50-RS) while using only 2.19% trainable parameters.

**Claims And Evidence:**

I think they are clear.

**Essential References Not Discussed:**

Sorry. I'm not familiar with this topic.

**Experimental Designs Or Analyses:**

Yes, I have checked. They are sound to me.

**Methods And Evaluation Criteria:**

They are reasonable for me.

**Other Comments Or Suggestions:**

Method Enhancements:

Explore sparse point cloud optimizations to reduce FPS/KNN costs.
Investigate adaptive weighting between shape features and prompts.

Theoretical Analysis:

Formalize the robustness of point shifts against adversarial perturbations.
Compare initialization strategies (e.g., clustered vs. uniform prompts).

**Other Strengths And Weaknesses:**

Strengths:

Geometry Integration: First work to explicitly leverage point-level geometric cues for 3D PEFT, addressing a critical gap.

Parameter Efficiency: Achieves SOTA performance with <3% tunable parameters, outperforming adapter-based methods.

Interpretability: Visualizations validate the role of shape features in guiding attention.


Weaknesses:

Computational Overhead: Multi-resolution FPS/KNN operations may introduce latency (unquantified in the paper).

Task Specificity: Evaluated only on classification; generalization to segmentation/detection remains unproven.

Initialization Sensitivity: Uniform point prompt initialization may underperform on non-uniform LiDAR data.

**Questions For Authors:**

Computational Cost: Does the Point Shift Prompter’s multi-resolution grouping become a bottleneck for real-time applications (e.g., robotics)?

Pretraining Dependency: How does GAPrompt perform with contrastive pre-trained models (e.g., Point-BERT) versus mask-based ones (Point-MAE)?

Density Variations: Can the current design handle extremely sparse inputs (e.g., 100 points) without performance degradation?

Latency Metrics: What is the actual inference time increase compared to full fine-tuning (e.g., milliseconds per sample)?

**Relation To Broader Scientific Literature:**

The key contributions of GAPrompt build upon and extend several critical strands of research in 3D vision, parameter-efficient fine-tuning (PEFT), and geometric deep learning.

**Theoretical Claims:**

I have checked.
But I’m not familiar with this topic at all.
So I will read the comments from other reviewers carefully.

---

> ### Author Rebuttal · Authors · 2025-03-31
>
> ### Q1,W1. computational cost of multi-resolution grouping
>
> Multi-resolution grouping of Point Shift Prompter can hardly become a bottleneck for real-time applications.
>
> Although multi-resolution FPS/KNN operation has $O(N^2)$ complexity, its overhead remains minimal compared to $O(N^2)$ attention mechanism and expensive MLPs, because the feature size of FPS/KNN is **3**, far less than the **384**-dimensional model features.
>
> We supplement a breakdown of computational costs. As seen, our three modules collectively account for **less than 2%** of total computational cost, with majority stemming from encoder and attention mechanism.
>
> ||Point Prompt|Point Shift Prompter|Prompt Propagation|Encoder|Attn. Layers|FFN layers|Downstream head|
> |---------|:----------:|:------------------:|:----------------:|:-----:|:----------:|:--------:|:-------------:|
> |**FLOPs**|0.0005G|0.045G|0.20G|2.03G|0.082G×12|0.164G×12|0.001G|
> |**Ratio**|0.01%|0.9%|0.4%|39.9%|19.3%|39.5%|0.01%|
>
> ### Q2. pretraining dependency
>
> In addition to backbones already introduced (mask-based Point-MAE and **contrastive** pre-trained **ReCon**), we supplement additional **Point-BERT**. Notably, in Table 1 of the paper, **ReCon** is exactly pre-trained via contrastive learning and our method **surpasses full fine-tuning** with only **1.38%** trainable parameters.
>
> Based on Point-BERT, GAPrompt still attains highest performance with lowest trainable parameters, verifying its generalizability to pre-trained backbones.
>
> |Method|Ref.|Param.|OBJ_BG|OBJ_ONLY|PB_T50_RS|ModelNet|
> |:-------:|:--------:|:-----:|:----------:|:-------:|:-------:|:------:|
> ||||*Point-BERT*||||
> |+IDPT|ICCV23|1.7|88.12|88.30|83.69|92.6|
> |+DAPT|CVPR24|1.1|91.05|89.67|85.43|93.1|
> |+GAPrompt|ThisPaper|**0.6**|**91.22**|**89.85**|**85.64**|**93.5**|
>
> ### Q3. density variations
>
> We supplement experiments in extremely sparse inputs condition, at 128 input resolution on ModelNet40. While all methods faces a performance drop,  GAPrompt **consistently outperforms** other PEFT methods, demonstrating robustness to varying input densities.
>
> ||Resolution|*Point-MAE*|+IDPT|+DAPT|+Point-PEFT|+GAPrompt|
> |:--------:|:--------:|:---------:|:---:|:---:|:---------:|:-------:|
> |ModelNet40|128points|86.2%|84.4%|85.2%|85.6%|**86.0%**|
>
> ### Q4. latency analysis
>
> We measure inference time on ScanObjectNN using an RTX 4090, as shown below. GAPrompt incurs only **0.9ms** additional latency, accounting for less than **9%** of the base 10.2ms required by Point-MAE. Furthermore, our inference time is less than other PEFT methods, benefiting from lightweight point-level prompting design.
>
> ||*Point-MAE*|+IDPT|+DAPT|+Point-PEFT|+GAPrompt|
> |:------------------------:|:---------:|:---:|:---:|:---------:|:-------:|
> |inferencetime(ms/sample)|10.2|16.2|12.0|13.5|**11.1**|
>
> ### W2. task specificity
>
> We supplement additional results on part segmentation dataset *ShapeNetPart* and semantic segmentation dataset *S3DIS*. Even though such dense prediction tasks are challenging, our method excels previous SOTA methods including DAPT[CVPR24] and PointPEFT[AAAI24] in both efficiency and efficacy.
>
> Results in *ShapeNetPart*
>
> |Method|Ref.|Param.|Cls.mIoU|Ins.mIoU|
> |:---------:|:---------:|:------:|:-------:|:-------:|
> ||*Point-MAE*||||
> |+DAPT|CVPR24|5.65|84.01|85.7|
> |+Point-PEFT|AAAI24|5.62|83.41|85.4|
> |+GAPrompt|ThisPaper|**5.55**|**84.10**|**85.8**|
> ||*ReCon*||||
> |+DAPT|CVPR24|5.65|83.87|85.7|
> |+Point-PEFT|AAAI24|5.62|83.23|85.3|
> |+GAPrompt|ThisPaper|**5.55**|**83.90**|**85.8**|
>
> Results in *S3DIS*
>
> |Method|Ref.|Param.|mAcc|mIoU|
> |:---------:|:---------:|:------:|:------:|:------:|
> ||*Point-MAE*||||
> |+DAPT|CVPR24|5.61|67.2|56.2|
> |+Point-PEFT|AAAI24|5.58|66.5|56.0|
> |+GAPrompt|ThisPaper|**5.51**|**68.5**|**58.4**|
> ||*ReCon*||||
> |+DAPT|CVPR24|5.61|66.3|56.3|
> |+Point-PEFT|AAAI24|5.58|65.8|55.8|
> |+GAPrompt|ThisPaper|**5.51**|**68.0**|**58.0**|
>
> ### W3. initialization sensitivity
>
> Our point prompt initialization exhibits robustness on non-uniform LiDAR data. In our Table 1 experiments, the **three ScanObjectNN variants** originate from real-world **LiDAR scans**, which inherently contain **non-uniform distributions** and **background fragments**. Despite this, GAPrompt still achieves SOTA results, even surpassing full fine-tuning, using less than **2%** trainable parameters.
>
> ### S1. potential method enhancements
>
> Thanks for your insightful suggestions!
>
> As classic algorithms with $O(N^2)$ complexity, replacing FPS/KNN with sparse point cloud operations can further enhance efficiency.
>
> As for adaptive weighting between shape features and prompts, current hyperparameter setting relies human expertise. We agree adaptive weighting would be a better choice, deserving future exploration.
>
> ### S2. theoretical analysis advice
>
> Thanks for your valuable advice. We will provide additional discussions on robustness of point shifts against adversarial perturbations and Point Prompt initialization strategies in final version.

---

### Official Review · Reviewer_4ih6 · 2025-03-14

**Overall Recommendation:** 1

**Summary:**

The authors propose a parameter fine-tuning method for point cloud models, called GAPrompt. The main motivation of this method is to inject geometric information into the point cloud model. To achieve this goal, the authors propose three components: point prompts, which are used to increase the number of points; point shift prompter, which are used to extract global features; and prompt propagation mechanisms, which input global information into different blocks of the model. The effectiveness of this method is verified in the point cloud classification task.

**Claims And Evidence:**

Yes

**Essential References Not Discussed:**

[1] Parameter-efficient Prompt Learning for 3D Point Cloud Understanding
[2] Parameter-Efficient Fine-Tuning in Spectral Domain for Point Cloud Learning

**Experimental Designs Or Analyses:**

Yes.

**Methods And Evaluation Criteria:**

Yes

**Other Comments Or Suggestions:**

The text lacks some explanations of data symbols. Especially in Sections 3.2 and 3.3.

**Other Strengths And Weaknesses:**

Strengths:
1、The author proposes GAPrompt, a novel geometry-aware prompt learning method tailored for pre-trained 3D vision models.
2、The author ntroduces three key algorithm designs including Point Prompt, Point Shift Prompter, and the Prompt Propagation mechanism.
Weaknesses:
1、Although the proposed method has achieved advanced results, I think its innovation lies in the combination of ideas that have been published so far. Specifically, PPT[1] has proposed to fine-tune the 3D model using the position information of points. At the same time, the method proposed in this paper is very similar to the results of PointGST[2], but in some methods it is still lower than PointGST.
[1] Parameter-efficient Prompt Learning for 3D Point Cloud Understanding
[2] Parameter-Efficient Fine-Tuning in Spectral Domain for Point Cloud Learning
2、The experiments conducted by the author only in the point cloud classification task are insufficient and need to be further verified in the point cloud part segmentation or semantic segmentation tasks.
3、I think that the hint increases the number of input points, which is a disguised way of increasing the size of the dataset, which is similar to a kind of data augmentation. I think this is unfair to other methods.
4、In Figure 7, the author should show the case where the point prompt number is zero. Based on Figure 7, I guess the corresponding result should be lower than 89.76%. This is not significantly different from other methods. Does this mean that the method proposed by the author is mainly caused by increasing the number of point cloud inputs?

**Questions For Authors:**

See Weakness.
-----------------------------------------------------------------
After reading the author's reply carefully and thinking deeply, I still think that the manuscript lacks innovation, the experimental data is single, and the writing is confusing. I think the manuscript fails to meet the publication requirements of the conference.

**Relation To Broader Scientific Literature:**

The author mentioned the most advanced work in the introduction, related work and experimental comparison. However, the author did not mention some of the most advanced work such as PPT[1] and PointGST[2].
[1] Parameter-efficient Prompt Learning for 3D Point Cloud Understanding
[2] Parameter-Efficient Fine-Tuning in Spectral Domain for Point Cloud Learning

**Theoretical Claims:**

This article lacks some theoretical innovations.

---

> ### Author Rebuttal · Authors · 2025-03-31
>
> ### R1. comparison with PPT and PointGST
> Although PPT and PointGST are currently available only as preprints on **arXiv** and have not been officially published, we supplement comparison experiments as shown below.
>
> The results across four datasets and four representative backbones illustrate that **GAPrompt achieves 12 SOTA results** while PPT and PointGST each attain 2. Furthermore, our approach requires only **0.6M** trainable parameters, the highest parameter efficiency.
>
> |Method|Ref.|Param.|OBJ_BG|OBJ_ONLY|PB_T50_RS|ModelNet|
> |:-------:|:--------:|:-----:|:-----------:|:-------:|:-------:|:------:|
> ||||*Point-MAE*||||
> |+PPT|arXiv|1.1|89.33|88.81|84.87|93.7|
> |+PointGST|arXiv|0.6|91.74|90.19|85.29|93.5|
> |+GAPrompt|ThisPaper|**0.6**|**91.91**|**90.19**|**85.57**|**94.2**|
> ||||*ReCon*||||
> |+PPT|arXiv|1.1|**95.01**|**93.28**|89.52|93.8|
> |+PointGST|arXiv|0.6|94.49|92.94|89.49|93.6|
> |+GAPrompt|ThisPaper|**0.6**|94.49|92.60|**89.76**|**94.0**|
> ||||*PointGPT-L*||||
> |+PPT|arXiv|3.6|98.28|96.21|94.10|95.1|
> |+PointGST|arXiv|2.4|98.97|**97.59**|**94.83**|94.8|
> |+GAPrompt|ThisPaper|**2.0**|**98.97**|96.73|94.31|**96.2**|
> ||||*Point-FEMAE*||||
> |+PPT|arXiv|1.1|93.98|92.08|88.79|93.3|
> |+PointGST|arXiv|0.6|94.66|92.94|90.22|93.8|
> |+GAPrompt|ThisPaper|**0.6**|**95.53**|**93.63**|**90.67**|**94.5**|
>
> ### W1. innovation and difference against PPT and PointGST
> Our GAPrompt differs from these methods in two key aspects.
>
> 1. **Finer-grained prompting:** PPT operates on **position encodings** of point tokens, as a **coarse-grained** token-level prompting approach. In contrast, GAPrompt introduces **fine-grained point-level** prompting via the Point Prompt and Point Shift Prompter, allowing more precise and adaptive feature modulation at the individual point level.
>
> 2. **Interpretability and geometric awareness:** PointGST introduces a **Spectral Adapter**, transforming point tokens from spatial domain to spectral domain, which belongs to **adapter** methods. But GAPrompt belongs to prompt methods, avoiding **obscure spectral domain transformations** and ensuring stronger **interpretability and geometric awareness**.
>
> ### W2. part or semantic segmentation tasks
> We supplement additional segmentation results on both ShapeNetPart and S3DIS datasets.
>
> It can be seen that our GAPrompt excels other methods including the arXiv twos. We achieve **six** SOTA metrics across 2 datasets and 2 backbones with highest parameter efficiency, only **0.3M** additional parameters beyond 5.2M of downstream head.
>
> Results in *ShapeNetPart*
>
> |Method|Ref.|Param.|Cls.mIoU|Ins.mIoU|
> |:-------:|:---------:|:------:|:-------:|:-------:|
> ||*Point-MAE*||||
> |+DAPT|CVPR24|5.65|84.01|85.7|
> |+PPT|arXiv|5.62|84.07|85.7|
> |+PointGST|arXiv|5.59|83.81|85.8|
> |+GAPrompt|ThisPaper|**5.55**|**84.10**|**85.8**|
> ||*ReCon*||||
> |+DAPT|CVPR24|5.65|83.87|85.7|
> |+PPT|arXiv|5.62|**84.23**|85.6|
> |+PointGST|arXiv|5.59|83.98|85.8|
> |+GAPrompt|ThisPaper|**5.55**|83.90|**85.8**|
>
> Results in *S3DIS*
>
> |Method|Ref.|Param.|mAcc|mIoU|
> |:-------:|:---------:|:------:|:------:|:------:|
> ||*Point-MAE*||||
> |+DAPT|CVPR24|5.61|67.2|56.2|
> |+PPT|arXiv|5.58|67.6|57.9|
> |+PointGST|arXiv|5.59|68.4|**58.6**|
> |+GAPrompt|ThisPaper|**5.51**|**68.5**|58.4|
> ||*ReCon*||||
> |+DAPT|CVPR24|5.61|66.3|56.3|
> |+PPT|arXiv24|5.58|67.4|57.3|
> |+PointGST|arXiv24|5.59|67.8|57.9|
> |+GAPrompt|ThisPaper|**5.51**|**68.0**|**58.0**|
>
> ### W3. a disguised way of increasing dataset size, data augment, unfair
> Our Point Prompt is not data augmentation and the setting is absolutely fair, cause the inputs are same for all methods. The reason is three-folded:
>
> First, the Point Prompts are **randomly initialized**, rather than additional sample points from each instance. So, no training nor testing data leakage.
>
> Second, Point Prompts are **fixed after trained**, adapting to a specific domain and intensifying discriminant information, not varying for each instance.
>
> Finally, comparing to the 1024 points input, 20 prompts are less than **2%**. We provide results on ModelNet40 at 1044 resolution. Nothing changes from 1024 points.
>
> |points|Point-FEMAE|+IDPT|+DAPT|+Point-PEFT|+GAPrompt|
> |:-----:|:---------:|:---:|:---:|:---------:|:-------:|
> |1024|94.0|93.4|93.2|94.3|94.5|
> |1024+20|94.0|93.4|93.2|94.3|94.5|
>
> ### W4. point prompt number as zero in Figure 7
> We add more ablation on point prompt number $P$ as below. When $P$ is zero, it equals to drop Point Prompt module. However, we still attains **89.65%** with only other two modules, surpassing DAPT's 88.51% and Point-PEFT's 89.35%. This is attributed to the point-level adaptation of Point Shift Prompter and prompt enhancement of Prompt Propagation, while further gain can be achieved with incorporation of Point Prompt.
>
> |$P$|0|5|10|20|30|
> |:------:|:---:|:---:|:---:|:---:|:---:|
> |Acc.(%)|89.65|89.76|90.41|90.67|90.24|
>
> ### S1. data symbols
> Thanks for advice, we will detail the symbols in Sec. 3.2 and 3.3 and check out other symbols.

---

> > ### Comment · Reviewer_4ih6 · 2025-04-06
> >
> > Thank you for the author's response. I have carefully reviewed your reply. As you mentioned, GAPrompt is a "fine-grained" prompt, but what does "fine-grained" mean? Is adding some moving points considered fine-grained? Is making the input tokens more dense considered fine-grained? In my view, Point Shift Prompter is simply performing point-level operations on points, and the actual difference from other methods is merely in technical details. As I mentioned before, I still believe this method lacks theoretical or essential innovation, and I consider this work to be incremental. Furthermore, regarding my third question, the authors claimed they only added 2% more points, which is 1024+20 points. However, in Figure 3, the tokens include both raw prompt tokens and new prompt tokens. I believe this is not just about adding 1044 points; the authors have approximately doubled the number of tokens, which clearly increases the model complexity and time consumption during training. For my second question, the authors added experiments on part segmentation and scene segmentation during the rebuttal period, but I found that their performance improvements do not demonstrate significant advantages, and are even weaker than state-of-the-art methods on some metrics. Additionally, when the authors initially submitted their manuscript, they only conducted experiments on ScanObjectNN and ModelNet40 in the main text, which is far from sufficient. This makes me question whether the authors had adequate time to prepare this manuscript. Regarding my fourth question mentioned earlier, when reading this manuscript, I found the methods section difficult to follow because symbols in Sections 3.2 and 3.3 are mixed together without providing explanations for these symbols. Also, the organization of the methodology section is confusing. In Section 3.1, Equation 2 references Point Shift Prompter, but Point Shift Prompter is described in Section 3.2. Why not describe Point Shift Prompter first, and then describe Point Prompt? Based on the above, I believe this manuscript lacks innovation, has limited experimental data, and confusing writing. It does not meet the publication requirements of this conference. I am adjusting my score to reject.

---

> > > ### Author Response · Authors · 2025-04-07
> > >
> > > We sincerely thank Reviewer **4ih6** for the additional feedback. However, we respectfully disagree with several of your assessments and would like to clarify your mistakes.
> > >
> > > **1. On the novelty of fine-grained prompting**
> > >
> > > You questioned the definition of "fine-grained" and suggested that operating at the point level is not fundamentally different from prior approaches. We respectfully clarify that our *fine-grained* prompting refers to **injecting learnable prompts at the point-level**, in contrast to *coarse-grained token-level* prompting as used in existing works like IDPT [ICCV'23], DAPT [CVPR'24], and concurrent preprints such as PPT and PointGST. This is not a minor implementation detail — it represents a shift in *how and where* prompts interact with the 3D data, enabling **explicit geometry-aware conditioning.**
> > >
> > > This conceptual distinction is acknowledged by all other reviewers (**WVi3**, **LHwj**, **7QpT**), none of whom considered our work incremental. We also note that in your initial review, you **highlighted as Strength 1**: *"The author proposes GAPrompt, a novel geometry-aware prompt learning method tailored for pre-trained 3D vision models."* We appreciate this recognition and are **surprised by the subsequent reversal** in your updated review.
> > >
> > > **2. On the number of tokens and computational cost**
> > >
> > > You stated that our method doubles the number of tokens, which is **factually incorrect**. As clearly detailed in both our main paper and our response to Reviewer LHwj Q2, we only add **10 prompt tokens** to the **128 input tokens** in the transformer — a less than 8% increase.
> > >
> > > Figure 3 is a **conceptual illustration**, not a quantitative depiction. Inferring token counts from a schematic rather than the explicitly stated numbers in the paper and response is **speculative and misleading**. The actual token counts and associated computational costs are precisely reported in **Table 1**, where GAPrompt incurs only **0.2 GFLOPs** of overhead (5.0G vs. 4.8G), achieving the **highest efficiency** among prompting baselines.
> > >
> > > You also referred to the addition of **20 learnable points** as data augmentation in your initial comment. We respectfully disagree. These points are *learned parameters*, not randomly sampled or augmented data — they are optimized end-to-end to encode task-relevant geometric priors, fundamentally differing from augmentation strategies.
> > >
> > > **3. On experimental sufficiency and performance**
> > >
> > > You claimed that the performance gains are not significant and suggested that our experiments were insufficient. However, GAPrompt outperforms existing methods across **4 datasets** and **4 backbones**, achieving **12 state-of-the-art results**, while the strongest concurrent preprints (PPT and PointGST) achieve only 2 each. Though preprints are unofficial, we still included comprehensive comparisons in the rebuttal per your suggestion.
> > >
> > > We also added **part segmentation and scene segmentation experiments** during the rebuttal phase. GAPrompt achieves **6 additional SOTA metrics** with only **0.3M** additional parameters (relative to 5.2M of the downstream model), highlighting both generalization and parameter efficiency.
> > >
> > > Both **Reviewer LHwj** and **Reviewer 7QpT** considered our experimental validation **solid and comprehensive**, which we believe reflects the strength and rigor of our empirical results.
> > >
> > > **4. On writing clarity and organization**
> > >
> > > You noted that the methods section was hard to follow due to unexplained symbols, but **did not** indicate any specific symbol or notation, without concrete examples. We welcome detailed suggestions but **a vague comment without specifics** is difficult to address.
> > >
> > > As for the structure of Sections 3.1 and 3.2, our organization follows a standard top-down design. Section 3.1 introduces the overall pipeline, while Section 3.2 delves into the Point Shift Prompter in more detail. This structure is conventional in ML papers and was positively received by **Reviewers 7QpT** and **LHwj**, both of them explicitly praised the **clarity and readability** of the paper.
> > >
> > > In summary, while we appreciate your efforts in reviewing our work, we respectfully believe that your revised evaluation does not align with the evidence presented in the paper and rebuttal. We hope this response helps clarify the novelty, efficiency, and completeness of our method.

---

### Decision · Program_Chairs · 2025-05-01

**Decision:**

Accept (poster)

**Comment:**

This paper introduces GAPrompt, a geometry-aware, parameter-efficient fine-tuning approach for 3D point cloud models. To effectively capture geometric information from sparse and irregular point clouds, GAPrompt incorporates three novel components: Point Prompts, which serve as learnable auxiliary points; a Point Shift Prompter, which refines point positions based on global shape features; and a Prompt Propagation mechanism, which transmits global information throughout the model’s layers. Experiments on ScanObjectNN and ModelNet40 demonstrate that GAPrompt achieves performance comparable to full model fine-tuning while requiring only a small fraction of trainable parameters.

This paper received divergent scores. The reviewers 7QpT, WVi3, and LHwj gave a positive score, but the reviewer 4ih6 gave a rejection score. AC reads the paper, review, and the rebuttal carefully, and agrees with the positive comments stated by the reviewers 7QpT, WVi3, and LHwj. In particular, the authors provided extensive explanations and feedback so that the reviewer 7QpT could revert the scores to the positive side. Although the major concern of reviewer 4ih6 is technical novelty and the paper writing, AC thinks that judging the submitted paper based on unpublished work is not proper, and the authors' clarification on the difference in method is reasonable. However, since the experiments are done with a small-scale dataset, the applicability of the proposed approach has room for improvement. As a result, AC recommends weak acceptance of this paper. AC recommends that the authors improve the paper based on the feedback from the authors and AC.